# The Activating Effect of Strong Acid for Pd-Catalyzed Directed C–H Activation by Concerted Metalation-Deprotonation Mechanism

**DOI:** 10.3390/molecules26134083

**Published:** 2021-07-04

**Authors:** Heming Jiang, Tian-Yu Sun

**Affiliations:** 1Lab of Computational Chemistry and Drug Design, State Key Laboratory of Chemical Oncogenomics, Peking University Shenzhen Graduate School, Shenzhen 518055, China; jiangheming@pku.edu.cn; 2Shenzhen Bay Laboratory, Shenzhen 518132, China

**Keywords:** directed C–H activation, concerted metalation–deprotonation mechanism, acid effect, electrophilicity, basicity

## Abstract

A computational study on the origin of the activating effect for Pd-catalyzed directed C–H activation by the concerted metalation-deprotonation (CMD) mechanism is conducted. DFT calculations indicate that strong acids can make Pd catalysts coordinate with directing groups (DGs) of the substrates more strongly and lower the C–H activation energy barrier. For the CMD mechanism, the electrophilicity of the Pd center and the basicity of the corresponding acid ligand for deprotonating the C–H bond are vital to the overall C–H activation energy barrier. Furthermore, this rule might disclose the role of some additives for C–H activation.

## 1. Introduction

Transition metal-catalyzed C–H activation to synthesize diverse organic molecules from simple hydrocarbon derivatives has emerged as a powerful tool for C–C and C-heteroatom bond formation and has received significant attention in recent years [1,2,3,4,5,6,7]. However, regioselectivity and reactivity have remained the most significant challenges in this active research field. Therefore, different strategies, especially directed C–H activation strategies, have been developed to improve regioselectivity and reactivity for C–H activation [8,9,10,11,12,13,14,15,16,17,18,19,20,21]. The directing groups (DGs) on substrates can chelate the metal catalyst and guide it to a specific position. Strong coordinating DGs usually contain strongly coordinating atoms, such as nitrogen, phosphorus, or sulfur atoms. They could widely promote C–H activation functionalization [22,23,24]. However, strong coordinating DGs are challenging to remove from the final products, limiting the utility of this strategy. Weak coordinating DGs (e.g., ketones, carboxylic acids, and ethers) are commonly occurring functional groups on the substrates and usually have much lower reactivity for C–H activation reaction [25,26,27,28,29,30]. Yu and co-workers have established several strategies to overcome the low reactivity of C–H activation by weak coordination, such as using counter cation effect, auxiliaries and monoprotected amino acid ligands [31,32,33,34,35]. Although impressive progress has been made, the scope of application of these strategies is still limited [36,37]. Using strong acid is also a widely used strategy to promote C–H activation; for example, palladium(II)-catalyzed ortho-selective C–H chlorination/bromination has demonstrated that proper strong acids (TFA, TfOH) could promote the reactivity of ortho-selective C–H bond cleavage (Scheme 1 and Scheme 2) [25,26,27,28,29,30,38,39,40]. These experimental results showed that higher catalytic activity occurred in reactions with lower pKa value acids.

The activating effect of a strong acid on Pd(II)-Catalyzed directed C–H activation has been known for a long time. Many kinds of C–H functionalization can be promoted by TFA or TfOH, such as carboxylation, [41] olefination, [42] arylation, [43,44] fluorination, [45] carbonylation, [46] trifluoromethylation, [47] amidation, [48] and oxygenation [49,50]. However, current understanding of the nature of strong acid-assisted C–H activation is still limited [51,52]. Fujiwara proposed that strong acid as a solvent facilitates the generation of highly cationic species([PdX]^+^) through ligand exchange (Scheme 3), which are very electrophilic. Cyclopalladium intermediates can be formed through the electrophilic aromatic substitution (S_E_Ar) of the C–H bond [53]. Other researchers have had a similar opinion to Fujiwara regarding the activating effect of strong acid [41,42,43,44,45,46,47,48,49,50].

Besides S_E_Ar, other frequently proposed mechanisms for C–H activation, including oxidative addition, σ-bond metathesis, and CMD mechanism, do not generate cationic species ([PdX]^+^) [54,55,56,57,58,59]. The CMD mechanism has been widely accepted as the best pathway for Pd-catalyzed C–H bond cleavage (Scheme 4) [60,61,62,63,64,65]. Therefore, we carried out density functional theory (DFT) [66] studies to explore the origin of the activating effect of strong acid on Pd-catalyzed directed C–H activation.

## 2. Results and Discussion

Since C–H activation is usually involved in the rate-determining step (RDS), [8,9,10,11,12,13,14,15,16,17,18,19,20,21,22,23,24,25,26,27,28,29,30,31,32,33,34,35,36,37,38,39,40,41,42,43,44,45,46,47,48,49,50] we hypothesized the relative free energy of transition states of C–H cleavage (ΔG_TS_^≠^) can determine the reactivity. The CMD mechanism was chosen for C–H activation, which usually has the lowest barrier among the frequently proposed mechanisms [57,58]. The transition state for the S_E_Ar mechanism could not be located (see Appendix A). Five substrates with different DGs were chosen; these substrates have previously been studied by experiments [38,39,40,49]. Trimeric [Pd(OAc)_2_]_3_ was chosen as the reference point of DFT calculations [67,68,69]. X-ray crystallography has provided evidence that when a strong acid such as TFA or TfOH is used, the OAc^−^ in the palladium acetate can be exchanged with TFA^−^ or OTf^−^ to form Pd(TFA)_2_ or Pd(OTf)_2_ [48,70,71,72]. As shown in Figure 1, for all of the five substrates, the ΔG_TS_^≠^ using three different Pd catalysts is in the same order: ΔG_TS_^≠^[Pd(OTf)_2_] < ΔG_TS_^≠^[Pd(TFA)_2_] < ΔG_TS_^≠^[Pd(OAc)_2_]. The order of reactivity is consistent with the experimental results, [38,39,40,73,74] indicating our DFT calculation is reliable.

Next, energy decomposition strategy was used to explore the origin of the activating effect for directed C–H activation by strong acid [63,75,76,77,78]. As shown in Scheme 5, ΔG_TS_^≠^ can be decomposed into two parts: ΔG_1_ and ΔG_2_^≠^. ΔG_1_ is the reaction energy caused by the coordination between the DG and the Pd catalyst. ΔG_2_^≠^ represents the energy needed to proceed with the C–H activation from int1. This strategy can reflect how the three different Pd catalysts influence ΔG_1_ and ΔG_2_^≠^, respectively.

For all of the five substrates, the order of ΔG_1_ using the three different Pd catalysts can be summarized as ΔG_1_[Pd(OTf)_2_] < ΔG_1_[Pd(TFA)_2_] < ΔG_1_[Pd(OAc)_2_] (see Figure 2a), which is in the reverse order of electrophilicity of the Pd catalysts: Pd(OTf)_2_ > Pd(TFA)_2_ > Pd(OAc)_2_ [48]. For ΔG_1_, the conclusion can be drawn that the more electrophilic Pd catalyst results in better coordination with DGs. For ΔG_2_^≠^, all of the five substrates have the consistent order: ΔG_2_^≠^[Pd(TFA)_2_] < ΔG_2_^≠^[Pd(OTf)_2_] < ΔG_2_^≠^[Pd(OAc)_2_] (see Figure 2b). The order of ΔG_2_^≠^ is different from that of ΔG_1_, and the C–H activation energy barrier of Pd(TFA)_2_ is the lowest. It was generally believed that a more electrophilic Pd catalyst would result in a lower barrier for the C–H activation step in the past [41,42,43,44,45,46,47,48,49,50]. However, our results do not support this belief: the electrophilicity of Pd(OTf)_2_ is strongest, but its C–H activation energy barrier is not the lowest. Therefore, the reason why the C–H activation energy barrier of Pd(TFA)_2_ is the lowest needs further study.

Inspection of the TS by the CMD mechanism demonstrated that ΔG_2_^≠^ is related to the metal center’s electrophilicity and the basicity of the acid ligand (Scheme 6a). To investigate the influence of electrophilicity and basicity on ΔG_2_^≠^ separately, an intermolecular model was built to decompose the effect of the two factors (Scheme 6b).

As shown in Figure 3, sub5 was chosen as an example for the intermolecular model study. In each row, the Pd catalyst in the three TSs is the same, but with three different external acid ligands, i.e., OAc^−^, TFA^−^ and OTf^−^. From each row, we can see how the basicity of acid ligands influences ΔG_2_^≠^. In each column, the external acid ligand in the three TSs is the same, but with three different Pd catalysts, i.e., Pd(OAc)_2_, Pd(TFA)_2_, Pd(OTf)_2_. From each column, we can see how electrophilicity of the Pd catalysts influences ΔG_2_^≠^. The three diagonal transition states, which have the same external acid ligand and the ligand of Pd catalysts, are most similar to our intramolecular mechanism.

For each row, the C–H activation energy barrier of the intermolecular model decreases with increasing basicity of the external acid ligand. The Pd(X)_2__OAc transition states have the lowest energy barrier. The energy differences between the OAc^−^ and TFA^−^ ligands are about 4.6 and 6.3 kcal/mol, and the similar gaps between the TFA^−^ and OTf^−^ ligands are about 5.6 and 6.8 kcal/mol. For each column, the C–H activation energy barrier of the intermolecular model decreases with the increasing electrophilicity of the Pd catalyst, and the Pd(OTf)_2__X transition states have the lowest energy barrier. The energy difference between the Pd(OAc)_2_ and Pd(TFA)_2_ catalysts is about 13.9~15.8 kcal/mol, much larger than the gap of 2.1~3.1 kcal/mol between the Pd(TFA)_2_ and Pd(OTf)_2_ catalysts. Therefore, the Pd(OTf)_2__OAc transition state with the strongest basicity of the OAc^−^ ligand and the strongest electrophilicity of the Pd(OTf)_2_ catalyst has the lowest energy barrier among the nine intermolecular models.

However, for the three diagonal transition states with the same external acid ligand and ligand of Pd catalysts, electrophilicity and basicity show an opposite trend. For example, although the electrophilicity of the metal center in Pd(OTf)_2_ is the strongest, the basicity of the acid ligand (OTf^−^) is the weakest. The Pd(TFA)_2__TFA transition state has the lowest energy barrier considering the influence of the external acid ligand’s basicity and the Pd catalyst’s electrophilicity, consistent with the lowest C–H activation energy barrier of Pd(TFA)_2_ in the intramolecular CMD process. According to the above discussion, it can be concluded that for the CMD mechanism, the electrophilicity of Pd catalysts and the basicity of acid ligands are critical to C–H activation.

Inspired by the lowest activation energy of Pd(OTf)_2__OAc, we hypothesized that C–H activation via an intermolecular CMD mechanism with a strong electrophilic Pd catalyst and strong external base may be favored, and some experiments support our hypothesis. In Yu’s work, the combination of Pd(OTf)_2_ and *N*-Methyl-2-pyrrolidone (NMP, a stronger base than TfO^−^) is crucial for C–H fluorination [45]. Our calculations indicate that the NMP-assistant intermolecular C–H activation process is about 9 kcal/mol lower in energy than the intramolecular C–H deprotonation by TfO^−^ (see Figure 4a). Buchwald and coworkers found that the combination of Pd(OAc)_2_/TFA and DMSO can improve the yield of C–H arylation [44]. They proposed that palladium black formation could be slowed by the addition of DMSO (10 mol%). Our calculations indicate that DMSO, a stronger base than TFA^−^, can also promote intermolecular deprotonation (see Figure 4b). As shown in Figure 4 and Appendix A, our calculations demonstrated that the intermolecular mechanism is more favorable than the intramolecular mechanism for the above two studies. Our findings might disclose the role of some additives for C–H functionalization.

## 3. Conclusions

In summary, the activating effect of strong acid for Pd(II)-catalyzed directed C–H functionalization was investigated with DFT calculations. Our results were consistent with previous experimental results and disclosed that the origin of the activating effect by strong acid comes from two parts: ΔG_1_ (coordination energy between the DG and the Pd catalyst) and ΔG_2_^≠^ (C–H activation energy). For the CMD mechanism, the electrophilicity of the Pd center and the basicity of the related acid ligand for deprotonation of the C–H bond is vital to the overall C–H activation energy barrier. This rule can be used to explain the role of some additives for C–H activation. It is hoped that our study could be used to improve the reactivity of some C–H functionalization reactions.

## Data Availability

More details about DFT calculation are in Appendix A. This material is available free of charge via the Internet at http://pubs.acs.org.

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
