# Peer review of "The Activating Effect of Strong Acid for Pd-Catalyzed Directed C–H Activation by Concerted Metalation-Deprotonation Mechanism"

_molecules, 2021, doi:10.3390/molecules26134083_

Round 1

Reviewer 1 Report

The manuscript reports an interesting approach to understand the origin of the activating effect for Pd-catalyzed directed CH activation by concerted metalation-deprotonation (CMD) mechanism, where generally strong acids are employed.

I found the DFT approach well done and the conclusions acceptable in one with the energy values obtained.

I found just an inaccuracy in the missing of subscripts in formulas reported in  equations 1 and 2 (page 1-2).  

Author Response

We thank the reviewer for his/her kind comments on our work. According to the reviewer’s suggestion, we have used subscripts in equations 1 and 2 (page 1-2), and  we have changed H2O, Pd(OAc)2, Na2S2O8, SO2NHR, CO2R to H2O, Pd(OAc)2, Na2S2O8, SO2NHR, CO2R in Revised Manuscript.   

Reviewer 2 Report

The paper of Jiang, H. et al. is important topic and can be accepted after minor revisions.

In reaction equations 1 and 2 subsscript numbers should be used: H2O, Pd((OAc)2, Na2S2O8. 

Row 49: (eq.3) is does not represent SEAr reaction, just a ligandum exchange.

Row 77: Scheme is "S" should be capital

Rows 180-308 The journal do not accept condensed references, as far as I know. Every reference should have a different number.

Author Response

The paper of Jiang, H. et al. is important topic and can be accepted after minor revisions.

Reply: We thank the reviewer for his/her kind comments on our work.

Point 1: In reaction equations 1 and 2 subsscript numbers should be used: H2O, Pd((OAc)2, Na2S2O8.

Response 1: We appreciate the reviewer’s suggestion. As the reviewer suggested, we have used subscripts in reaction equations 1 and 2. We have changed H2O, Pd(OAc)2, Na2S2O8, SO2NHR, CO2R to H2O, Pd(OAc)2, Na2S2O8, SO2NHR, CO2R in Revised Manuscript. 

Point 2: Row 49: (eq.3) is does not represent SEAr reaction, just a ligandum exchange.

Response 2: As the reviewer suggested, we have revised the corresponding sentences. We have changed " Fujiwara proposed that strong acid as solvent facilitates the generation of highly cationic species ([PdX]+), and it is very electrophilic. The cyclopalladium intermediates can be formed through the electrophilic aromatic substitution (SEAr) of the C-H bond (eq 3)" to " Fujiwara proposed that strong acid as solvent facilitates the generation of highly cationic species ([PdX]+) through ligand exchange (eq 3), and it is very electrophilic. The cyclopalladium intermediates can be formed through the electrophilic aromatic substitution (SEAr) of the C-H bond" in Revised Manuscript. 

Point 3: Row 77: Scheme is "S" should be capital

Response 3: As the reviewer suggested, we have changed "scheme" to "Scheme" in Revised Manuscript. 

Point 4: Rows 180-308 The journal do not accept condensed references, as far as I know. Every reference should have a different number.

Response 4: As the reviewer suggested, we have assigned a different number to each reference, and updated the corresponding number in the text in Revised Manuscript.